# Images as Tables: In-Context Learning with TabPFN for Low-Data Detection of AI-Generated Images

**Jan Philip Walter** [* 1]  **Shashank Agnihotri** [* 1]  **Margret Keuper** [1 2]

## Abstract

AI-generated image detection is a moving-target problem: detectors trained on one generator often fail when a new generator appears, and only a few labeled examples are available. We study a simple image-to-table formulation for this regime, where each image is encoded by a frozen DINOv3 backbone, its CLS feature is reduced to a 500-dimensional structured row with PCA, and TabPFN performs real/fake classification by in-context tabular inference rather than task-specific classifier training. This turns fake-image detection into low-data structured prediction over learned visual features, making detector adaptation depend on the labeled context set instead of gradient-based fine-tuning. On GenImage, LATTE, a recent state-of-the-art detector, remains stronger when many labeled samples from all generators are available, by 7.4% in the largest pooled setting, but DINOv3-PCA-TabPFN is stronger in the practically important low-data regime, outperforming LATTE by up to 8.2%, and in transfer settings where the detector must generalize from one generator to another. These results position tabular foundation models as a strong complementary adaptation mechanism for image forensics, shifting adaptation from detector retraining to lightweight in-context updates with a small labeled set of examples. Code hyperlink here.

## 1. Introduction

Photorealistic image generators have made image authenticity a moving-target classification problem (Keuper & Keuper, 2025). A detector that performs well on im-

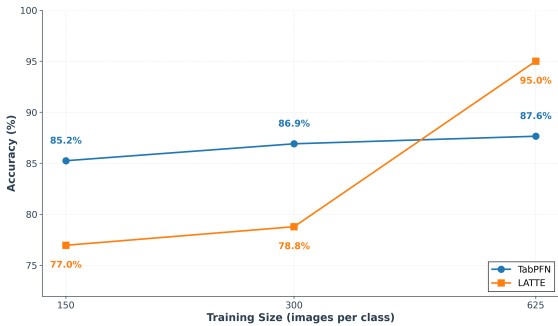

*Figure 1.* In pooled generator setting, LATTE reaches the highest accuracy (real/fake class detection) at the largest training size, but DINOv3-PCA-TabPFN is stronger at the smaller shared training sizes, the regime targeted by our in-context detector adaptation.

ages from known generators may fail after the generator architecture, training data, sampling procedure, or post-processing pipeline changes. This is the central difficulty in AI-generated image detection: the label boundary is not only real versus generated, but also generator-dependent. Prior work has therefore emphasized cross-generator evaluation, dataset shortcuts, and generator fingerprints as key obstacles for reliable synthetic-image detection (Yu et al., 2019; Epstein et al., 2023; Cozzolino et al., 2024; Yermakov et al., 2026; Pei et al., 2026).

Most recent detectors remain image-domain systems. They either train visual classifier heads on real/fake data, use CNN or ViT backbones, analyze frequency artifacts, or exploit diffusion-specific traces (He et al., 2016; Dosovitskiy et al., 2021; Liu et al., 2022; Vasilcoiu et al., 2025; Mahara & Rishe, 2026). This paper asks a deliberately different question: once an image has been converted into a strong frozen representation, can the final forensic decision be treated as structured-data inference? As shown in Figure 1, this gives a lightweight detector-adaptation route: the image encoder remains fixed, while the TabPFN context changes with the small labeled set available for the new setting.

We evaluate DINOv3-PCA-TabPFN, a three-stage detector. First, a frozen DINOv3 ViT-B/16 model maps each image to a 768-dimensional CLS representation (Siméoni et al., 2025). Second, PCA maps the representation to 500 dimensions, matching the feature limit of the TabPFN version used in this study. Third, TabPFN performs binary classification

[1]Machine Learning Group, University of Mannheim, Mannheim, Germany [2]Max-Planck-Institute for Informatics, Saarland Informatics Campus, Germany. Correspondence to: Shashank Agnihotri <shashank.agnihotri@uni-mannheim.de>.

*Accepted as Spotlight Oral at the $2^{nd}$ FMSD Workshop at the International Conference on Machine Learning*, Seoul, South Korea. 2026. Copyright 2026 by the author(s).

over the resulting table by in-context tabular inference (Hollmann et al., 2023). This decouples representation learning from detector adaptation: new labeled samples are added to the TabPFN context rather than used to optimize a new image classifier.

We make the following main contributions: First, we formulate AI-generated image detection as tabular inference over frozen visual features. Second, we evaluate this formulation on GenImage across generator-aware protocols that separate pooled performance, per-generator difficulty, single-generator transfer, and pairwise specialization (Zhu et al., 2023). Lastly, we compare against LATTE and identify a clear empirical trade-off: TabPFN is stronger in low-data and several cross-generator regimes, while LATTE reaches the highest peak when many labeled samples from all generators are available.

## 2. Related Work

**AI-generated image detection.** Modern synthetic images are produced by both GAN-based and diffusion-based generators, with text-conditioned diffusion systems now forming a major part of the detection landscape (Goodfellow et al., 2014; Brock et al., 2019; Dhariwal & Nichol, 2021; Rombach et al., 2022; Nichol et al., 2022; Gu et al., 2022). Related evaluation work also studies broader properties of text-to-image generation, such as geographic diversity (Basu et al., 2026); our focus is complementary, namely detector adaptation once images have been generated. Detection methods are commonly grouped into image-space, frequency-space, and data-driven approaches, with generalization beyond the generator distribution seen during training being a central challenge (Pei et al., 2026; Epstein et al., 2023; Yermakov et al., 2026). Early work showed that generated images can contain generator-specific fingerprints, but this also implies that detectors may overfit to model-specific artifacts rather than learn a generator-agnostic real/fake boundary (Yu et al., 2019; Cozzolino et al., 2024). GenImage (Zhu et al., 2023) was introduced as a large-scale benchmark for this cross-generator problem, containing real ImageNet images paired with fake images from eight generators. We use this setting as it directly tests whether a detector adapts when the generator changes.

**Specialized forensic detectors.** Recent detectors improve robustness by designing image-domain objectives or exploiting traces left by the generation process. Frequency-domain cues are also relevant because generated-image methods can explicitly shape spectral distributions (Jung & Keuper, 2021). For example, CLIP-based detectors use strong pre-trained visual-language representations for AI-generated image detection (Cozzolino et al., 2024), while LATTE uses latent trajectory embeddings from diffusion denoising and is therefore a strong diffusion-oriented baseline (Vasilcoiu

*Table 1.* Generator-aware evaluation protocols. All splits are disjoint and class-balanced.

| Protocol | Train | Test | What it measures |
|---|---|---|---|
| Multi-Multi | 8 generators | 8 generators | Pooled detection when all generators are represented in training and testing. |
| Multi-Single | 8 generators | 1 generator | Per-generator difficulty after broad training coverage. |
| Single-Multi | 1 generator | 8 generators | Transfer from one observed generator to the full generator set. |
| Single-Single | 1 generator | 1 generator | Pairwise generator specialization and transfer to every other generator. |

et al., 2025). These methods remain specialized image-forensic systems: adaptation typically involves training or fitting a detector for the target data regime. In contrast, instead of replacing such detectors, we test whether the final real/fake decision can be shifted to a low-data in-context inference problem once images have been converted into structured feature rows.

**Frozen visual representations.** Pretrained visual backbones are widely used as feature extractors in computer vision and forensics. CNN-based models such as ResNet and ConvNeXt provide strong image representations, while Vision Transformers model images as token sequences and can capture global dependencies through self-attention (He et al., 2016; Liu et al., 2022; Dosovitskiy et al., 2021). Self-supervised visual foundation models further strengthen this feature-extraction view: DINOv2 and DINOv3 provide general-purpose representations without task-specific fine-tuning (Oquab et al., 2024; Siméoni et al., 2025). Since training procedures can affect how vision models use their learned representations (Gavrikov et al., 2024), we treat the frozen encoder choice as an empirical part of the pipeline. Our method follows this direction but uses the frozen visual model only to create the table; the forensic classifier itself is a tabular foundation model.

**Tabular foundation models.** TabPFN is a prior-data fitted network that performs small-data tabular classification by approximating Bayesian-style inference from a labeled context set, avoiding dataset-specific gradient-based training at prediction time (Hollmann et al., 2023). Recent extensions improve the accuracy and scale of tabular foundation models (Hollmann et al., 2025; Grinsztajn et al., 2025). In-context learning has also been explored for specialized sensing settings such as THz imaging with VLMs (Poggi et al., 2025). Our work uses TabPFN outside its native tabular domain by treating each image embedding as one structured row. This positions AI-generated image detection as a multimodal structured-data problem with TabPFN supplying the low-data, in-context decision rule.

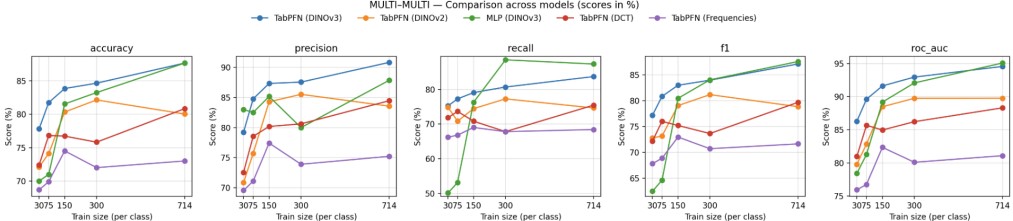

*Figure 2.* Representation and classifier comparison in the Multi-Multi development setting. DINOv3 features with TabPFN give the strongest and most stable performance across accuracy, precision, recall, F1, and ROC-AUC, while DINOv2, frequency features, and the MLP baseline are weaker in the low-data regime. All compared encoders and settings are explained in Appendix Section C.

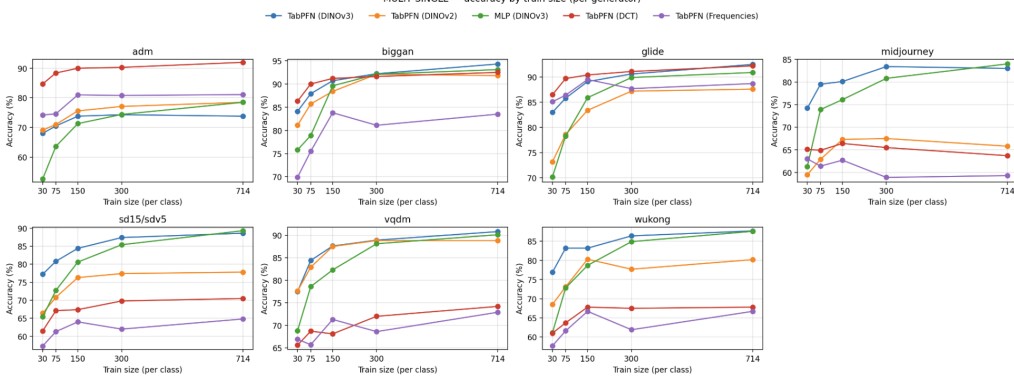

*Figure 3.* Multi-Single accuracy by training size and test generator. Training uses all fake generators, while each panel evaluates one test generator, highlighting which generators remain difficult even after broad training coverage.

## 3. Method

Given an image $x$, DINOv3 produces a frozen feature vector $h(x) \in \mathbb{R}^{768}$. Images are loaded as RGB images, resized to a shorter edge of 256 pixels, center-cropped to $224 \times 224$. The backbone is kept in evaluation mode and no image-domain fine-tuning is performed. The feature matrix is then reduced with Incremental PCA (Ross et al., 2008) to obtain $z(h(x)) \in \mathbb{R}^{500}$. The reduction is fitted on the training features and applied to the corresponding test features.

TabPFN receives the reduced feature rows and their binary labels, where 0 denotes real and 1 denotes generated. In contrast to a learned detector head, the classifier is not optimized on the target split. It performs prediction through the prior-data fitted network mechanism, which approximates Bayesian-style tabular inference from the labeled context (Hollmann et al., 2023). The practical consequence is that adaptation is moved from gradient-based detector retraining to changing the labeled context set. The TabPFN variant used here is limited to 10,000 samples and 500 features.

## 4. Experimental Setup

We use GenImage, which contains ImageNet-derived real images and fake images from eight generators: ADM, Big-GAN, GLIDE, Midjourney, Stable Diffusion v1.4, Stable Diffusion v1.5, VQDM, and wukong (Zhu et al., 2023). The

set includes both diffusion generators (Dhariwal & Nichol, 2021; Nichol et al., 2022; Rombach et al., 2022; Gu et al., 2022) and the GAN-based BigGAN (Goodfellow et al., 2014; Brock et al., 2019). The full generator counts are reported in Table 3. We focus on cross-generator classification rather than degraded-image robustness.

**Protocols and sample sizes.** Table 1 summarizes the four settings. Training uses $k \in \{25, 30, 75, 150, 300, 625\}$ fake samples per generator and the same number of real samples. In Multi training, $k = 625$ already yields 5,000 fake and 5,000 real images, matching TabPFN's 10,000-sample limit. Every test set contains 10,000 samples. Multi testing uses 5,000 real images and 625 fake images per generator; Single testing uses 5,000 real images and 5,000 fake images from one generator. LATTE is evaluated under the same split logic for $k \in \{150, 300, 625\}$, because smaller configurations were not supported by the available setup.

## 5. Results

**Which image-to-table representation works?** Figure 2 compares several representations and classifiers in the Multi-Multi development setting. The DINOv3-TabPFN combination is the strongest and most stable option across accuracy, precision, recall, F1, and ROC-AUC. DINOv2-TabPFN is consistently weaker, and hand-crafted frequency features

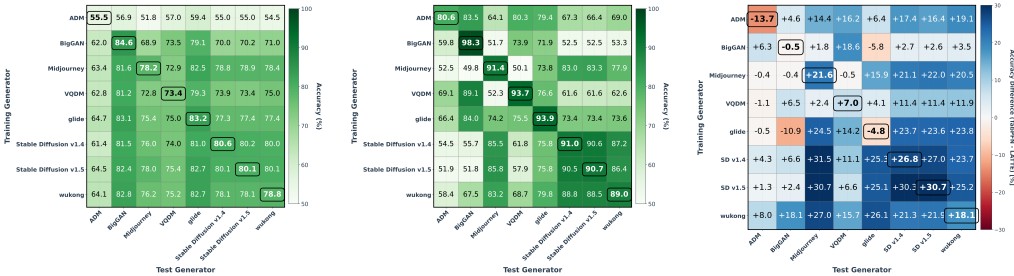

*Figure 4.* Pairwise generator transfer. Left and middle: TabPFN accuracy matrices at $k = 25$ and $k = 625$. Right: TabPFN minus LATTE at $k = 625$, where positive values indicate a TabPFN advantage.

do not close the gap. A small MLP trained on the same DINOv3 features becomes competitive with more data, but TabPFN is stronger in the small-context regime. This is the key methodological result: the gain is not simply "use a vision backbone", but the combination of a strong frozen visual representation with a tabular prior that works well from few labeled context examples. Additional details on metrics and encodings are given in Section B, Section C, and supporting plots are included in Section D.2.

**Which generators remain difficult?** Figure 3 evaluates the Multi-Single setting, where training uses all generators but testing is restricted to one generator at a time. This setting isolates per-generator difficulty after broad training coverage. DINOv3-PCA-TabPFN remains the most reliable option across generators: BigGAN and GLIDE become easy with few examples, while ADM, Midjourney, and wukong remain harder and benefit more gradually from additional context. This covers the generator-wise robustness behavior that is hidden by pooled accuracy. The same trend appears in the PCA diagnostics in Figures 11 and 12: easier generators show clearer real/fake separation in the frozen DINOv3 feature space, while harder generators overlap more strongly with real images.

**When is the method better than LATTE?** The central comparison is shown in Figure 1. In the pooled setting, LATTE reaches the best high-data peak and is 7.4% above DINOv3-PCA-TabPFN at $k = 625$. However, DINOv3-PCA-TabPFN is better at the smaller shared training sizes, with a maximum gain of 8.2%. This supports the intended use case of the paper: fast detector adaptation when only a small labeled context set is available. The full pooled context-growth curve is reported in Figure 5, and the complete Multi-Single and Single-Multi LATTE comparisons are shown in Figures 16 and 17.

**What happens under pairwise generator transfer?** Figure 4 shows the most fine-grained generalization setting, where training and testing are restricted to generator pairs. With only $k = 25$ samples, the TabPFN matrix is comparatively balanced. With $k = 625$, same-generator accuracies often exceed 90%, but some off-diagonal transfer decreases, indicating stronger specialization to generator-specific cues.

*Table 2.* Main empirical takeaways. DINOv3-PCA-TabPFN is advantageous in low-data and cross-generator transfer, while LATTE reaches the highest pooled peak at the largest training size.

| Setting | Observation |
|---|---|
| Pooled low-data | DINOv3-PCA-TabPFN reaches 78% accuracy already at $k = 25$ and outperforms LATTE by up to 8.2% at the smaller shared training sizes. |
| Pooled high-data | LATTE becomes stronger at $k = 625$, exceeding DINOv3-PCA-TabPFN by 7.4% in the largest pooled setting. |
| Per-generator tests | Generator difficulty varies substantially; BigGAN and GLIDE are easier, while ADM and Midjourney are harder. Full diagnostic curves are in the appendix. |
| Pairwise transfer | At $k = 625$, DINOv3-PCA-TabPFN is better than LATTE in 54/64 train-test generator pairs, with gains up to 31.5%. |

In direct comparison to LATTE at $k = 625$, DINOv3-PCA-TabPFN is better in 54 of 64 train-test pairs, with gains up to 31.5%. This is the strongest evidence that the method is not only a pooled low-data detector, but also a useful transfer mechanism between generator families.

## 6. Discussion and Conclusion

The evidence supports a focused but useful claim. Tabular foundation models can serve as efficient downstream decision modules for image forensics when images are first converted into structured rows by a strong vision foundation model. This matters when adapting to a generator with only a small labeled set: changing the TabPFN context is simpler than training a detector-specific head, while retaining competitive cross-generator behavior. More broadly, this follows reliability and generalization benchmarking work showing that robustness should be evaluated under structured shifts rather than only under i.i.d. test conditions (Agnihotri et al., 2025d;a;c; Oei et al., 2026).

The study also points to a broader evaluation direction for structured foundation models. In addition to native tabular datasets, these models can be tested on structured representations induced from other modalities. AI-generated image detection is a meaningful stress test because the real/fake boundary shifts across generators and because the amount of verified labeled data may be small in practical adaptations.

# Acknowledgements

The authors acknowledge support by the state of Baden-Württemberg through bwHPC. S.A. & M.K. acknowledge support by DFG Research Unit 5336 - Learning to Sense (L2S). The authors gratefully acknowledge the computing time provided on the high-performance computer HoreKa by the National High-Performance Computing Center at KIT (NHR@KIT). This center is jointly supported by the Federal Ministry of Education and Research and the Ministry of Science, Research, and the Arts of Baden-Württemberg, as part of the National High-Performance Computing (NHR) joint funding program (https://www.nhr-verein.de/en/our-partners). HoreKa is partly funded by the German Research Foundation (DFG). Additionally, we thank Steven Adriaensen for his helpful discussion and feedback.

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

# Supplementary Material

## Appendix Contents

## A. Implementation and Dataset Details

This appendix provides reproducibility details and supporting diagnostics for the main paper. The organization follows the main text: we first describe the feature extraction and tabularization pipeline, then define the evaluation metrics, then detail the alternative image encodings and baselines, and finally provide additional plots for pooled scaling, representation diagnostics, per-generator difficulty, and cross-generator transfer.

Anonymized code for the work is available at: https://github.com/jpwalter30/Towards-Generalizable-Detection-of-AI-Generated-Images.

### A.1. Feature extraction and tabularization

The feature extraction code iterates over the GenImage folder structure and stores image paths, binary labels, generator names, and split identifiers. Each image is loaded with PIL, converted to RGB, resized such that the shorter edge is 256 pixels, center-cropped to $224 \times 224$, converted to a tensor, and normalized with ImageNet statistics. DINOv3 is used in evaluation mode and produces a CLS-token feature matrix with dimensionality $N \times 768$. IncrementalPCA is fitted on the training features and maps these embeddings to $N \times 500$ feature rows, matching the feature limit of the TabPFN version used here. The evaluation script constructs balanced train and test sets for the four generator-aware protocols described in Table 1.

### A.2. Dataset composition

Table 3 reports the generator-level image counts used in the GenImage cross-generator study. The benchmark contains seven diffusion-based generators and one GAN-based generator, BigGAN. This mixture is important for interpreting the transfer results, because training on BigGAN and testing on diffusion models is not the same shift as transferring between diffusion generators.

*Table 3.* GenImage generator composition used for the cross-generator study.

| Generator | Train | Test | Total |
|---|---|---|---|
| ADM | 162k | 6k | 168k |
| BigGAN | 162k | 6k | 168k |
| GLIDE | 162k | 6k | 168k |
| Midjourney | 162k | 6k | 168k |
| Stable Diffusion v1.4 | 162k | 6k | 168k |
| Stable Diffusion v1.5 | 166k | 8k | 174k |
| VQDM | 162k | 6k | 168k |
| wukong | 162k | 6k | 168k |

## B. Evaluation Metrics

We report accuracy, precision, recall, F1, and ROC-AUC for binary real/fake detection. The generated class is treated as the positive class. Let TP, TN, FP, and FN denote true positives, true negatives, false positives, and false negatives.

### B.1. Accuracy

Accuracy measures the overall fraction of correctly classified images:

$$\text{Accuracy} = \frac{\text{TP} + \text{TN}}{\text{TP} + \text{TN} + \text{FP} + \text{FN}}.$$

### B.2. Precision

Precision measures how often images predicted as generated are actually generated:

$$\text{Precision} = \frac{\text{TP}}{\text{TP} + \text{FP}}.$$

### B.3. Recall

Recall measures how many generated images are detected:

$$\text{Recall} = \frac{\text{TP}}{\text{TP} + \text{FN}}.$$

### B.4. F1

F1 is the harmonic mean of precision and recall:

$$\text{F1} = \frac{2 \cdot \text{Precision} \cdot \text{Recall}}{\text{Precision} + \text{Recall}}.$$

### B.5. ROC-AUC

ROC-AUC summarizes threshold-independent ranking quality. It is the area under the receiver operating characteristic curve, which plots the true-positive rate against the false-positive rate as the decision threshold changes. This is useful because two methods can have similar accuracy at one operating point while separating real and generated samples differently across thresholds.

## C. Encodings and Baselines

The main method uses DINOv3 features, PCA, and TabPFN. The ablations change either the image-to-table encoding or the downstream classifier to test whether the observed gains come from the visual representation, the tabular prior, or hand-crafted forensic statistics.

### C.1. DINOv3 + TabPFN

This is the proposed configuration. A frozen DINOv3 ViT-B/16 backbone extracts one CLS embedding per image, PCA reduces the representation to 500 dimensions, and TabPFN performs in-context binary classification over the resulting feature rows.

### C.2. DINOv2 + TabPFN

This follows the same pipeline as DINOv3 + TabPFN, but uses DINOv2 as the frozen feature extractor. This tests whether the result is specific to the newer DINOv3 representation or also holds for an earlier self-supervised visual backbone.

### C.3. MLP on DINOv3 features

The MLP baseline uses the same DINOv3 visual features as the proposed method, but replaces TabPFN with a small gradient-trained classifier. The architecture is a multi-layer perceptron with hidden dimensions $512$ and $128$, dropout, and a binary output layer. This baseline tests whether TabPFN's low-data behavior is better than simply training a lightweight neural classifier on the same frozen representation.

### C.4. TabPFN with DCT features

The DCT encoding uses a discrete cosine transform inspired by JPEG-style frequency analysis. Images are split into $8 \times 8$ blocks, an orthonormal 2D DCT is computed for each block, and summary statistics of the DCT coefficients are used as tabular features. In particular, coefficient-wise statistics across blocks summarize local frequency content, and the DC component can be omitted to focus on texture rather than global brightness.

### C.5. TabPFN with frequency features

The frequency encoding uses a two-dimensional Fast Fourier Transform (FFT2) to represent images in the frequency domain. The spectrum is divided into radial bins from low to high frequencies, and magnitude and phase statistics are extracted per bin. This produces a compact tabular representation for TabPFN and tests whether generic frequency artifacts are sufficient compared to learned DINO features.

These comparisons should therefore not be read as TabPFN architecture variants. In most cases TabPFN is fixed; what changes is the encoding that turns images into structured rows. The core question is which representation makes in-context tabular inference effective for fake-image detection.

## D. Additional Results and Diagnostics

The main paper keeps the figures needed for the central argument. This section provides the supporting plots for context-size scaling, representation choice, generator difficulty, and cross-generator transfer.

### D.1. Pooled Low-Data Scaling

Figure 5 shows the standalone scaling behavior of TabPFN in the pooled Multi-Multi setting. The main paper uses Figure 1 for the direct LATTE comparison; this appendix plot isolates TabPFN and shows that accuracy increases smoothly with more labeled context examples.

### D.2. Representation Diagnostics

Figures 6 and 7 provide additional evidence for the representation choice. Across metrics, DINOv3 features paired with TabPFN are more stable than DINOv2 features, hand-crafted frequency features, and the MLP baseline in the smallest regimes. Frequency features improve with additional data, but remain weaker than the learned visual representation.

Figures 8 to 10 show complementary PCA views of the frozen DINOv3 feature space. These plots are diagnostic rather than final evaluation results: they show that the visual representation already contains both real/fake structure and generator-dependent structure before the TabPFN decision step.

### D.3. Per-Generator Difficulty

Figures 11 to 13 expand the per-generator analysis. Big-GAN and GLIDE are generally easier because their fake samples are more separated from real samples in the DINOv3-PCA projection, while ADM and Midjourney are more difficult because their features overlap more strongly with real images. The generator-wise curves show that this is not only a pooled effect: the same generators remain consistently easier or harder across training sizes.

Figures 14 and 15 further compare encodings and model families in the Multi-Single setting. They support the same conclusion as the main paper: generator difficulty persists across classifier choices, but DINOv3 features with TabPFN remain the most reliable low-data configuration. These plots are shown as separate figures to improve readability.

### D.4. Cross-Generator Transfer

Figures 16 and 17 give the full cross-generator comparison to LATTE. Multi-Single asks whether a detector trained with broad generator coverage works on each individual test generator. Single-Multi is stricter: it asks whether a detector trained on one fake generator transfers to the full generator set. These plots support the main-paper conclusion that the TabPFN context is especially useful in transfer-heavy settings, although the exact difficulty depends strongly on which generator is observed during training.

Figures 18 and 19 provide additional Single-Multi comparisons across encodings and baselines. Splitting these plots into separate figures makes the individual panels more readable than the earlier grouped appendix layout.

### D.5. Compact Summary Plot

Figure 20 provides a compact grouped-bar summary of TabPFN behavior across the evaluated settings. It is included as a quick visual reference for the appendix, while the main paper reports the corresponding conclusions in Table 2.

## E. Limitations and Future Work

This study focuses on cross-generator classification and does not evaluate degraded inputs such as JPEG compression, blur, or low resolution. The PCA extraction is deliberately simple; future work could use stronger compression or feature-selection methods. Future work should also evaluate newer TabPFN variants with larger context and feature limits (Hollmann et al., 2025; Grinsztajn et al., 2025). The study also does not address model-internal safety or safety-pretraining mechanisms for generative models (Agnihotri et al., 2025b); it evaluates the downstream forensic detection problem. The method is a complementary low-data detector, not a replacement for specialized forensic models.

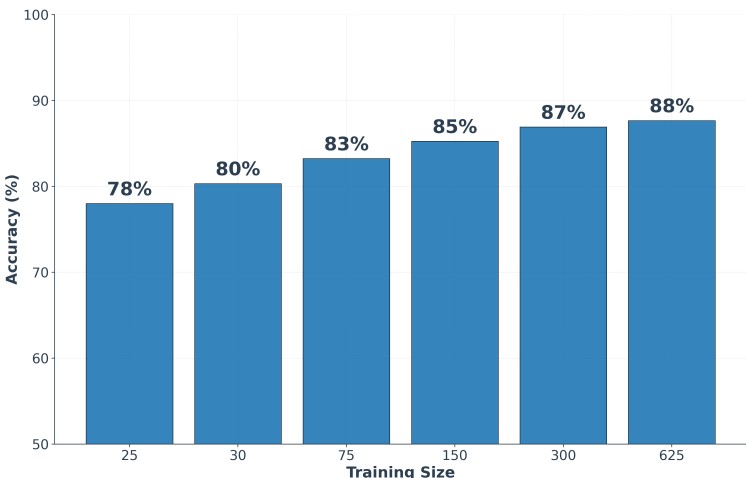

*Figure 5.* Multi-Multi context-size scaling for TabPFN. Accuracy increases steadily with more labeled context examples, but the main paper focuses on the direct comparison to LATTE rather than this standalone scaling curve.

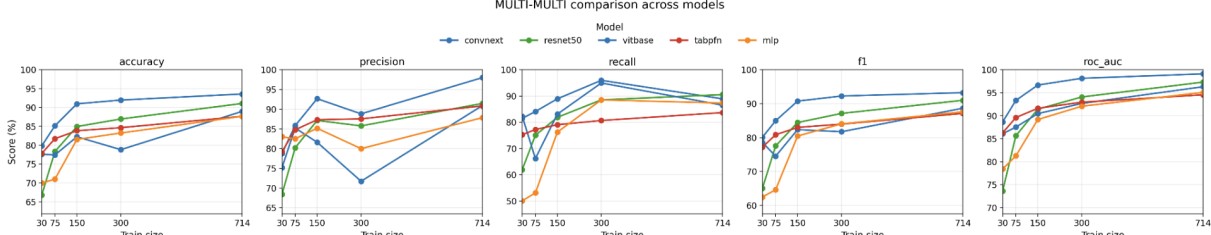

*Figure 6.* Additional Multi-Multi comparison across model families. DINOv3 with TabPFN is the most stable option across the tested sample sizes, while DINOv2, frequency features, and the MLP baseline are less reliable in the smallest regimes.

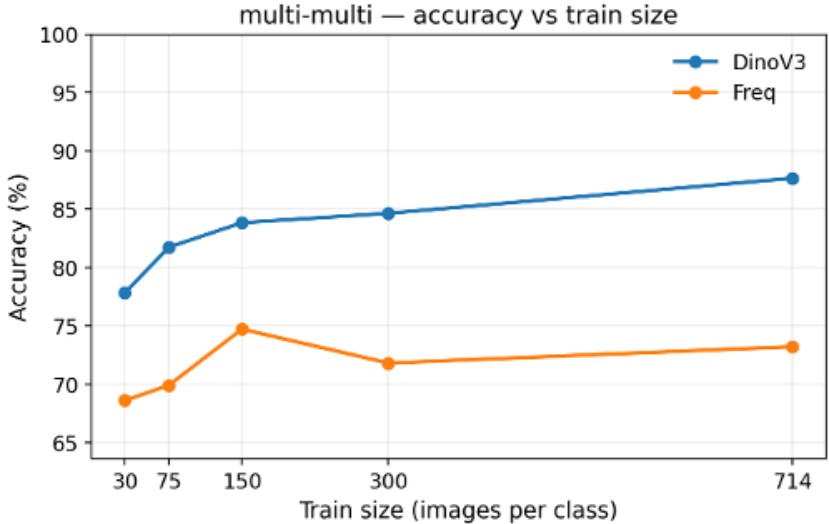

*Figure 7.* Frequency-domain feature comparison. Frequency features improve with training size, but remain weaker than DINOv3 features with TabPFN, supporting learned visual embeddings rather than hand-crafted frequency statistics alone.

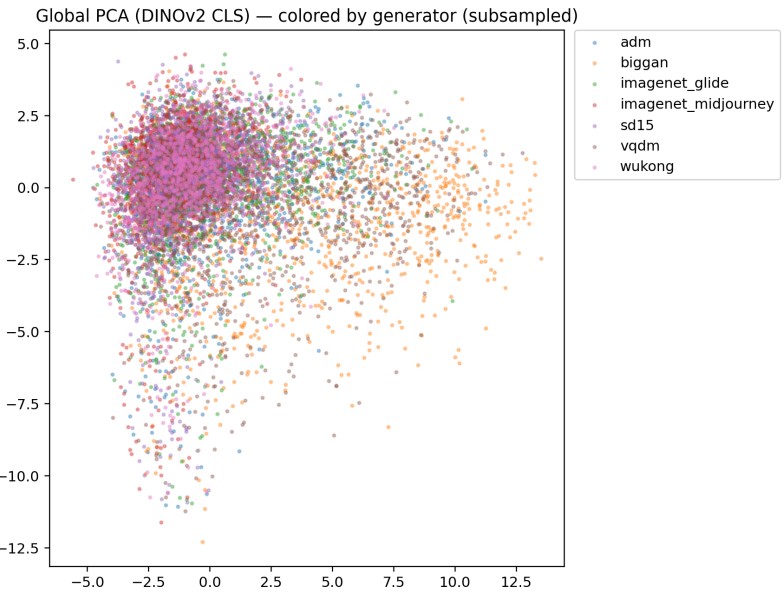

*Figure 8.* Global PCA projection of DINOv3 CLS features colored by generator. The plot shows that the frozen representation contains generator-dependent structure before the TabPFN decision step.

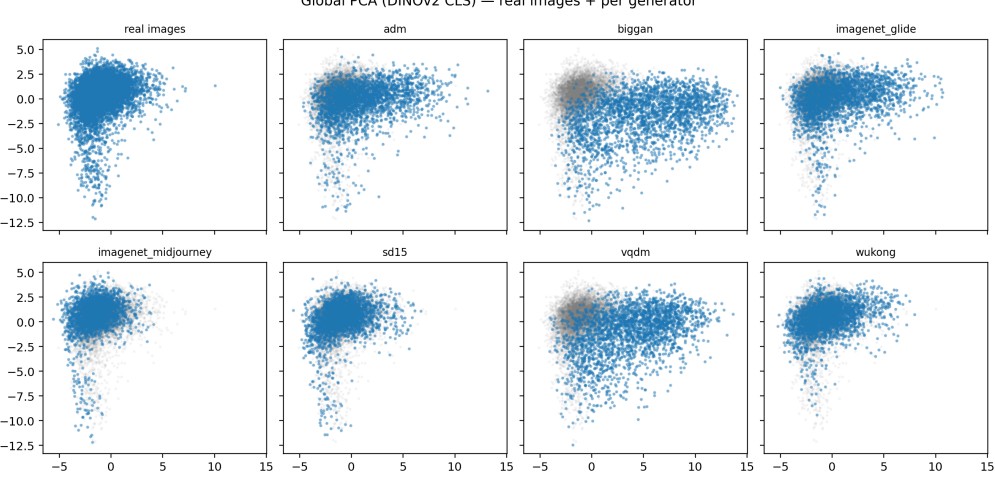

*Figure 9.* Generator-wise PCA facets for real and generated images. The facets make visible that some generators are more separated from real images than others, which explains part of the Multi-Single difficulty pattern.

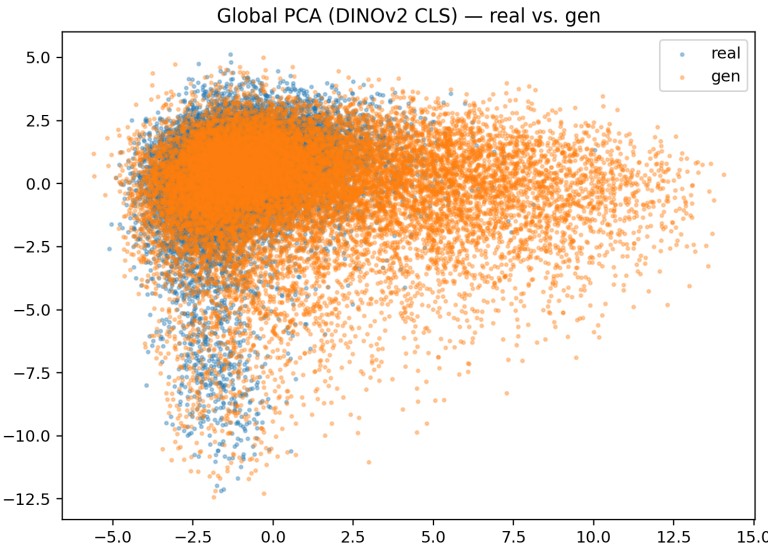

*Figure 10.* Global PCA projection of real versus generated images. The overlap indicates that a simple linear projection is not enough for perfect separation, but the visible structure supports lightweight downstream classification.

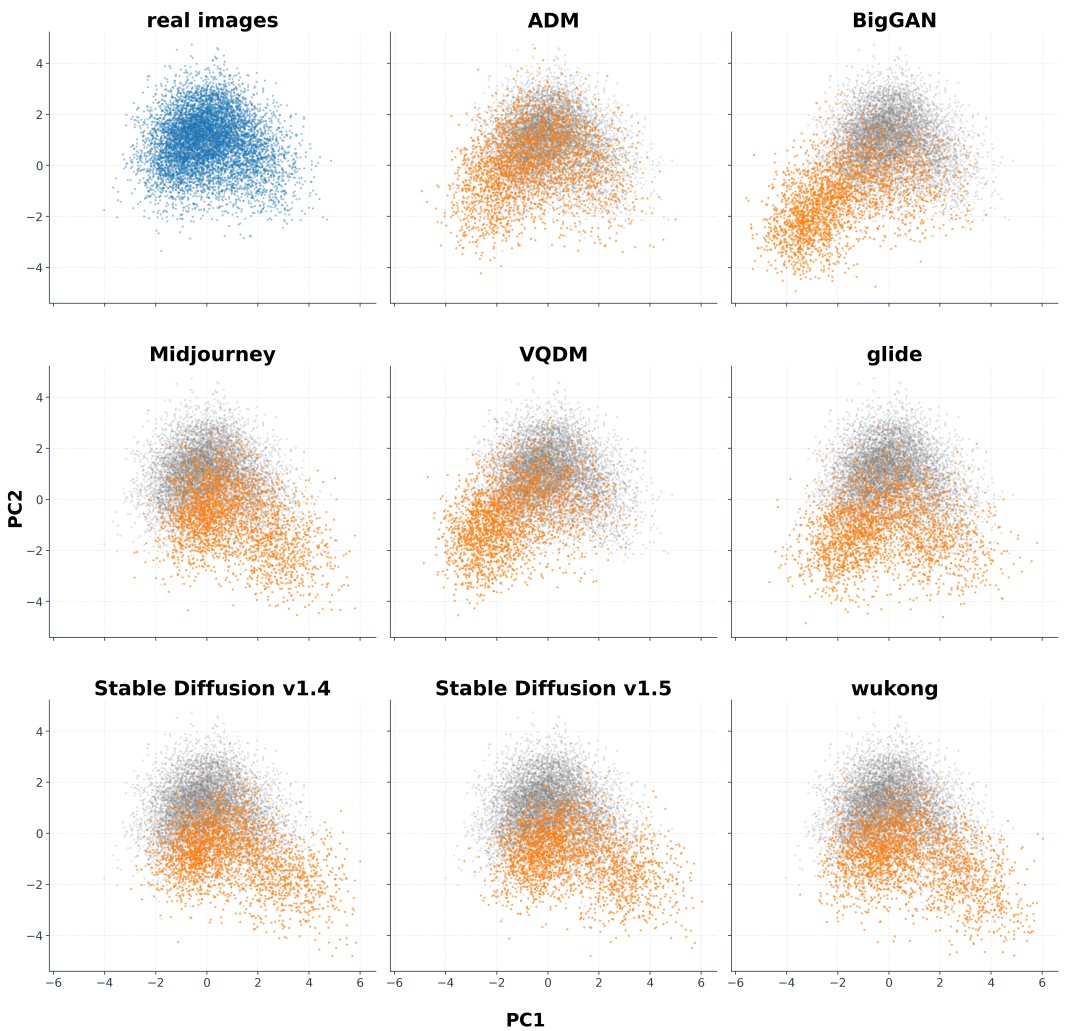

*Figure 11.* PCA grid by generator. BigGAN and GLIDE show clearer real/fake separation, while ADM and Midjourney overlap more strongly with real images.

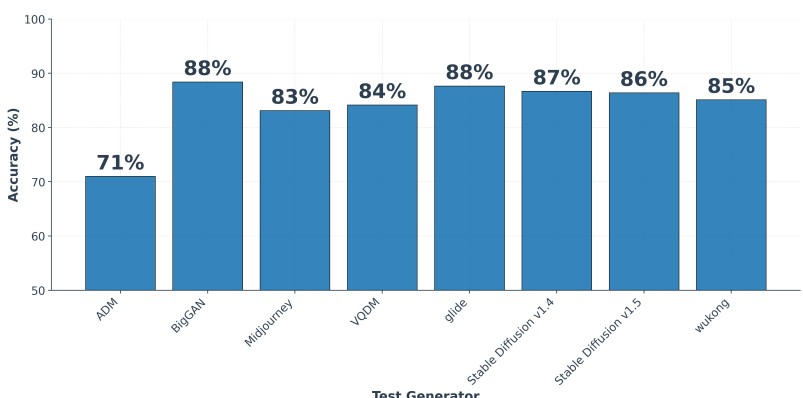

*Figure 12.* TabPFN Multi-Single accuracy by test generator. This compact view shows that generator difficulty varies substantially even when training uses all generators.

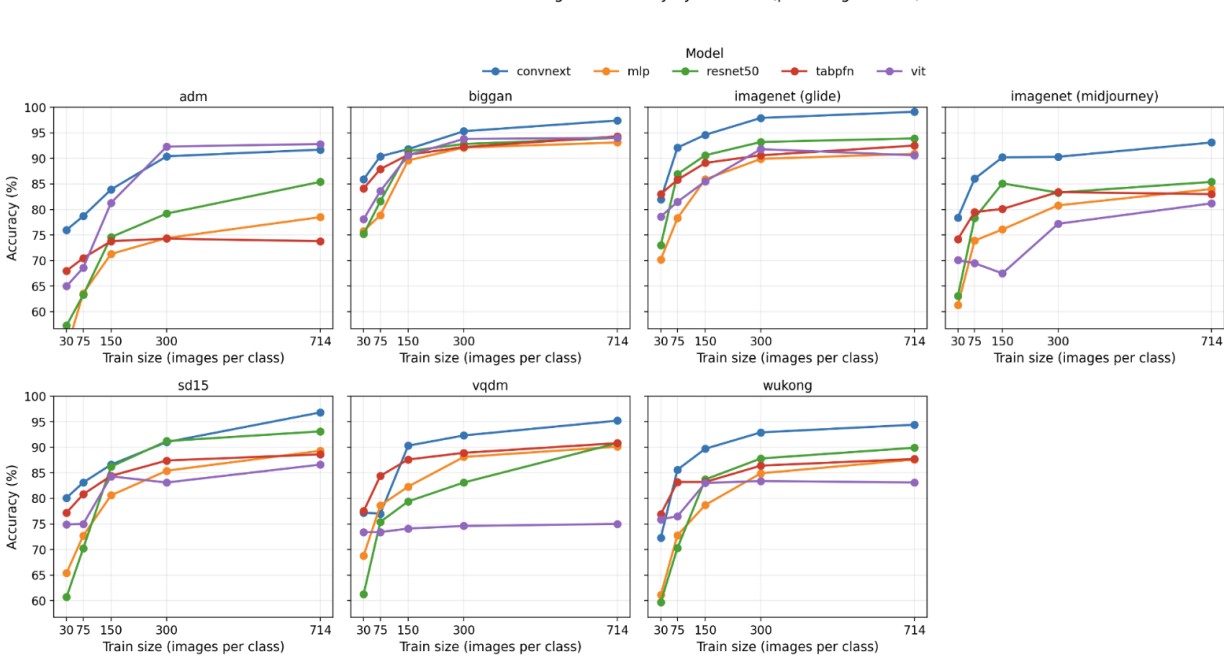

*Figure 13.* Multi-Single accuracy by training size and test generator. This plot complements the main-paper Multi-Single figure with the full per-test-generator training-size curves.

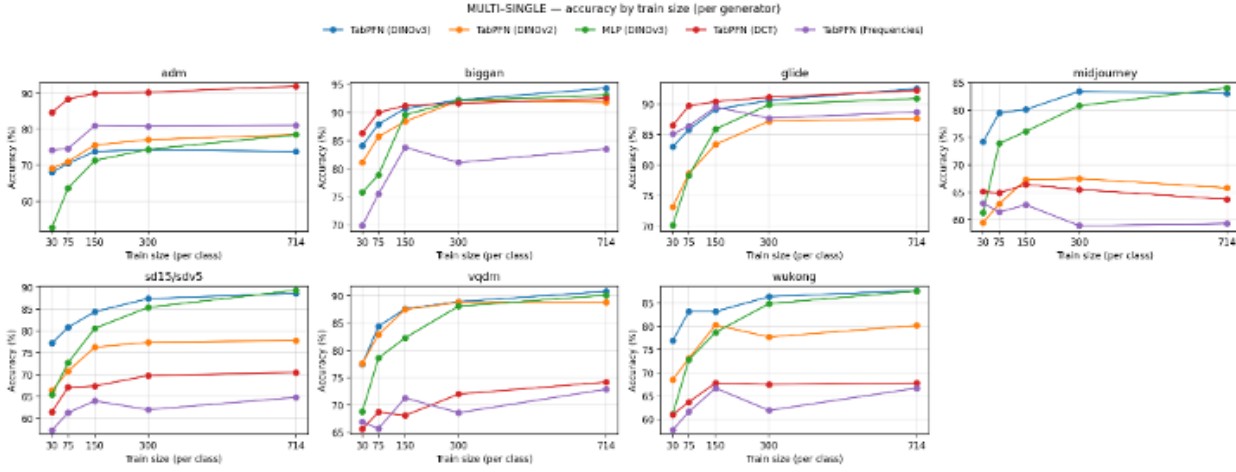

*Figure 14.* Multi-Single comparison across TabPFN input encodings. The results show that generator difficulty persists across encodings, but DINOv3 features remain the most reliable low-data representation.

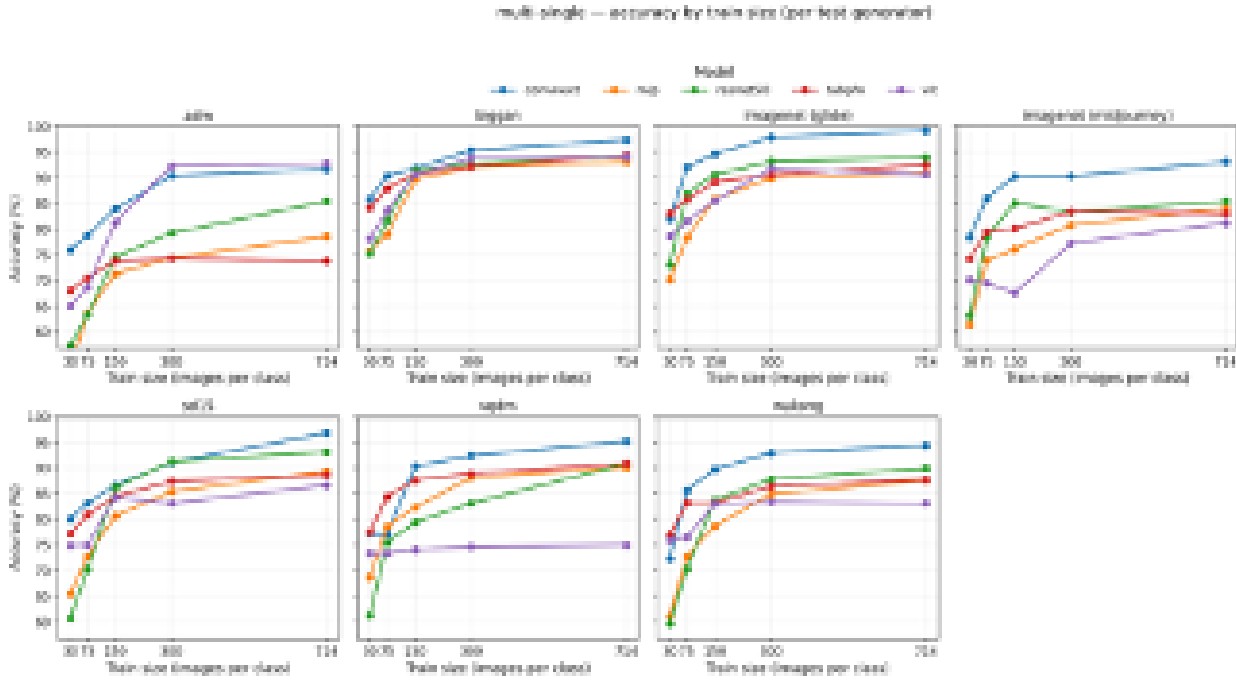

*Figure 15.* Multi-Single comparison across model families. The plot compares the proposed image-to-table pipeline against alternative classifiers and shows that difficult generators remain challenging across model choices.

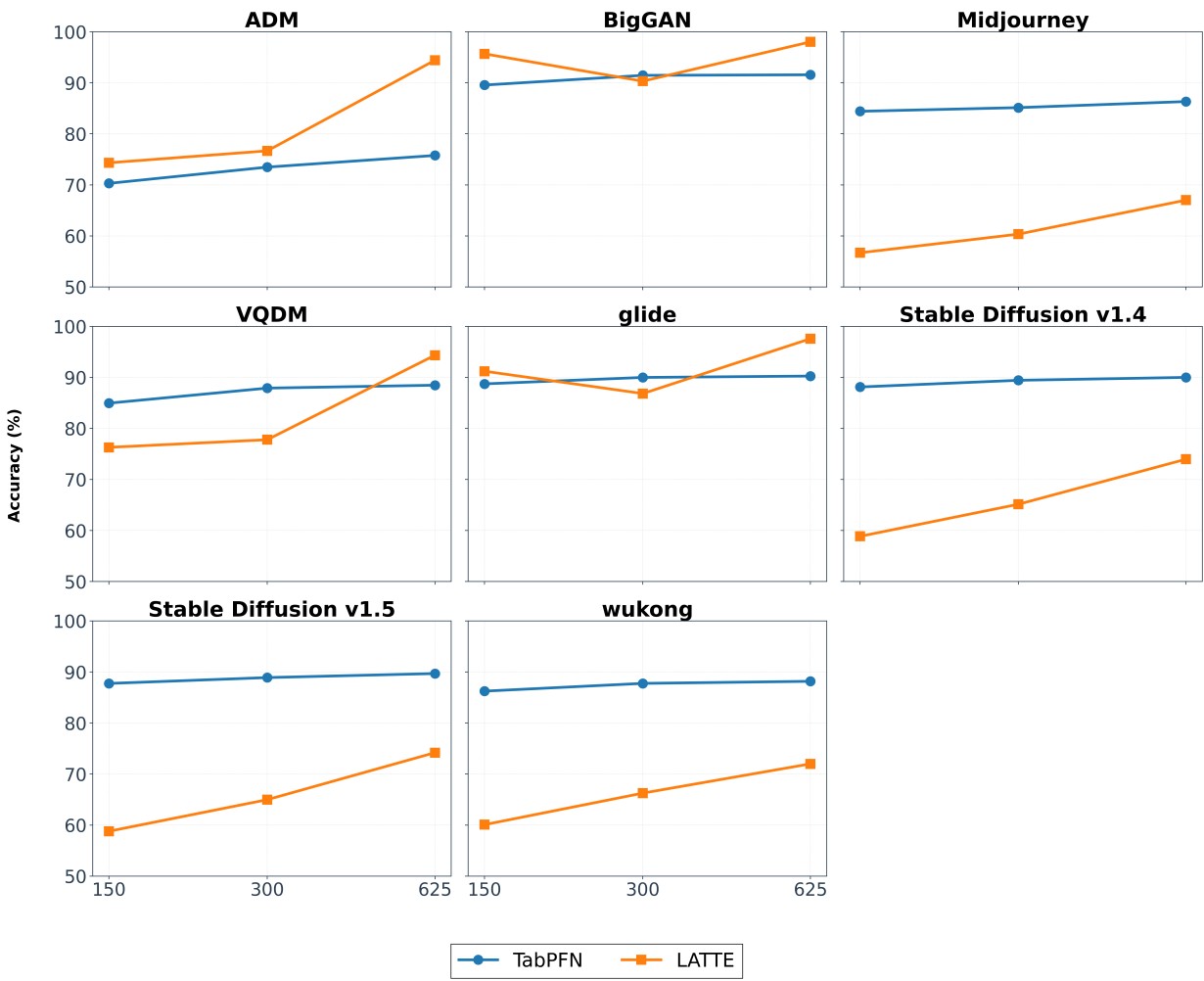

*Figure 16.* Multi-Single comparison to LATTE. The detector is trained with broad generator coverage and evaluated on one test generator at a time.

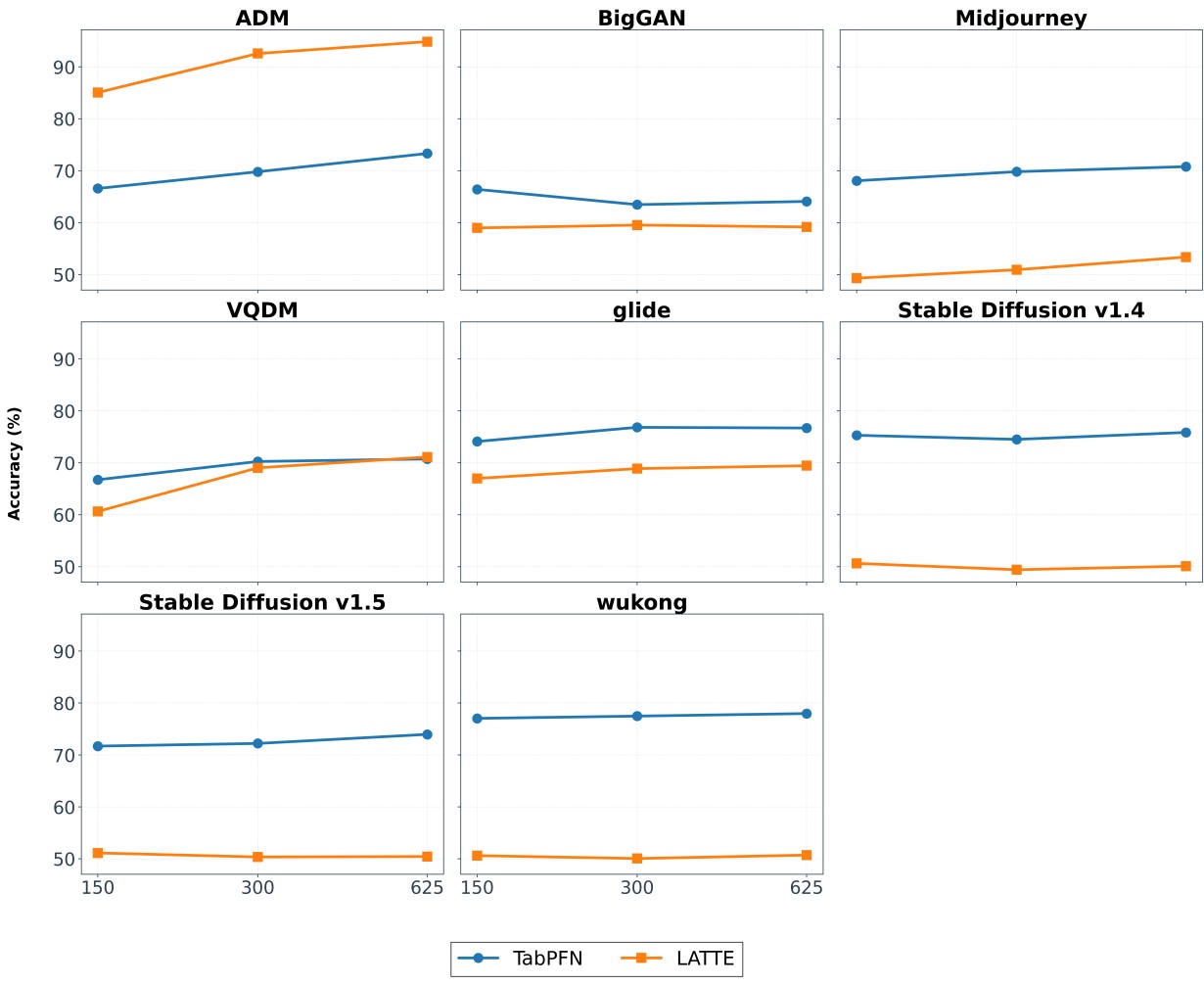

*Figure 17.* Single-Multi comparison to LATTE. The detector is trained on one fake generator and evaluated on the full generator set, making this a stricter cross-generator transfer setting.

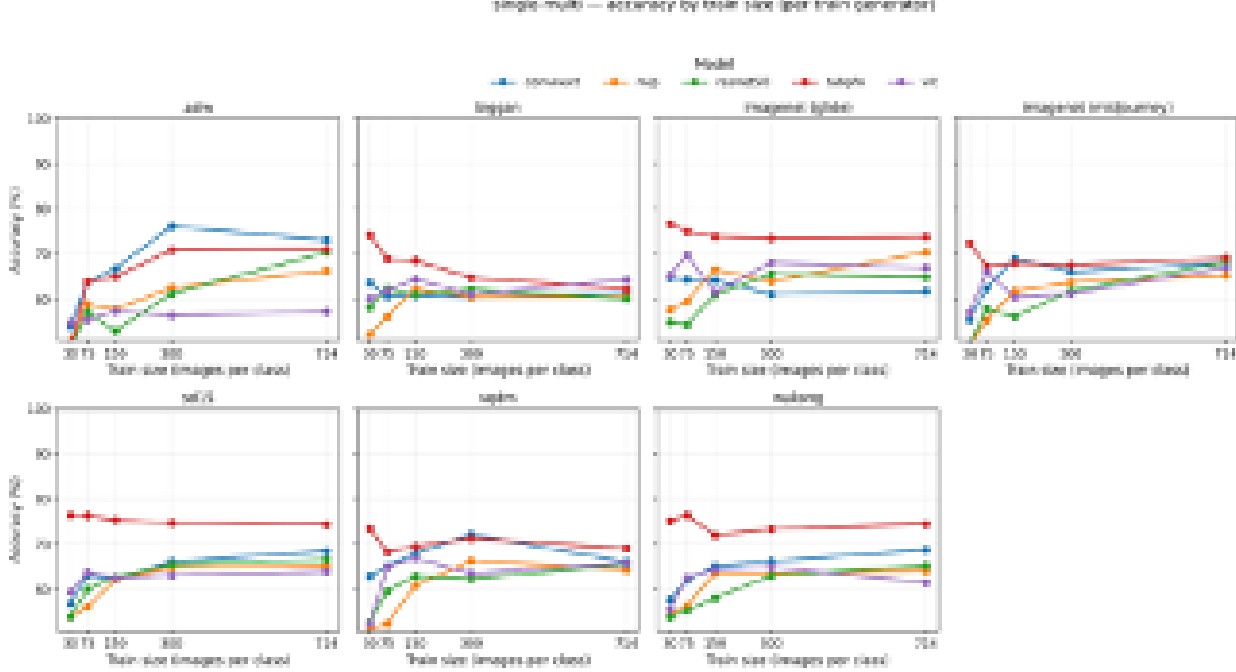

*Figure 18.* Single-Multi comparison across model families. This setting is stricter than Multi-Single because only one fake generator is observed during training and the detector is evaluated on the full generator set.

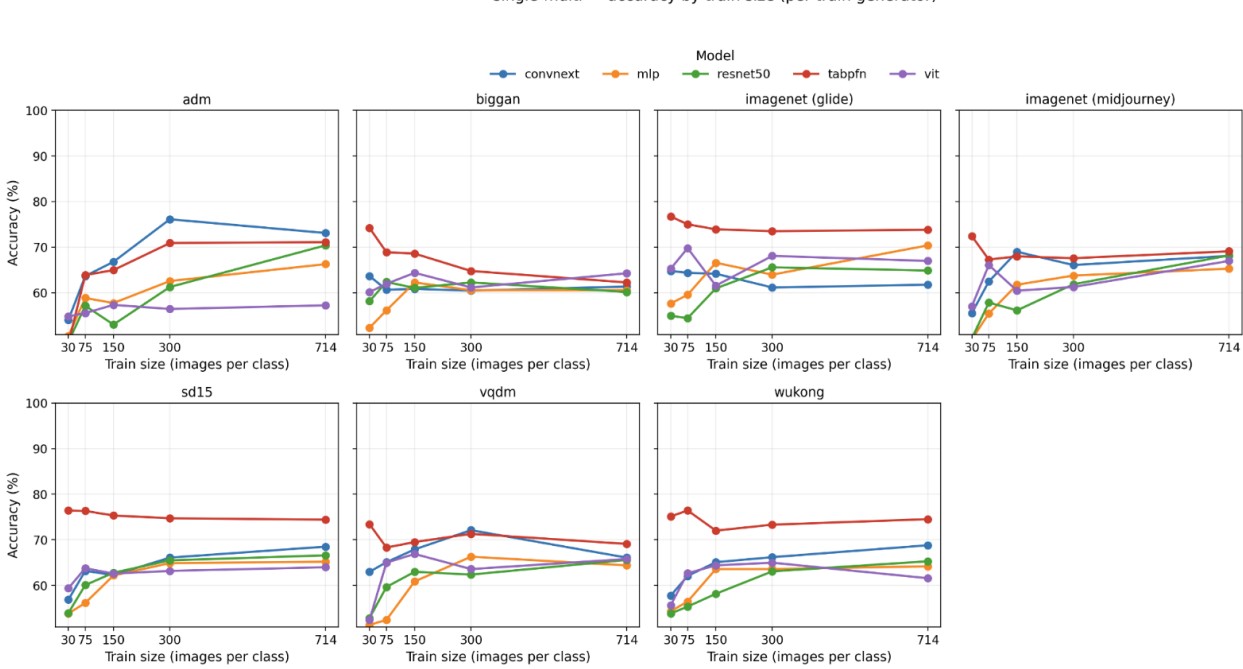

*Figure 19.* Single-Multi baseline comparison. The plot provides the baseline trends for transfer from one observed fake generator to the full generator set.

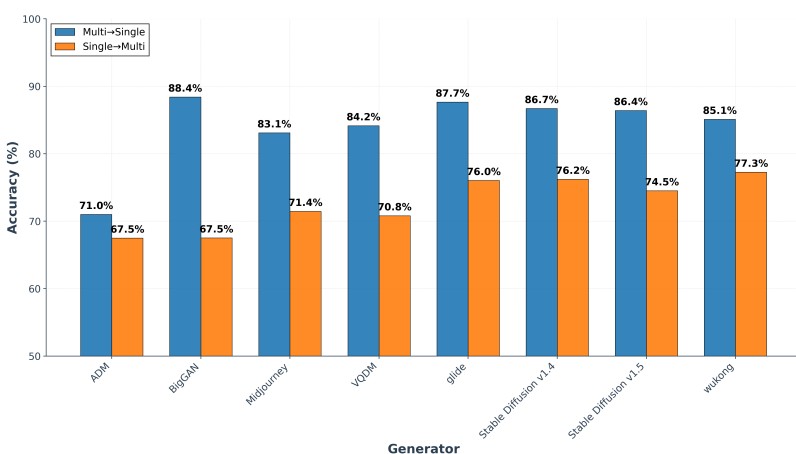

*Figure 20.* Grouped TabPFN result summary across generator-aware settings. This figure provides a compact visual overview of the main TabPFN trends discussed in the paper and appendix.

