# OpenReview forum: "Images as Tables: In-Context Learning with TabPFN for Low-Data Detection of AI-Generated Images"
_ICML.cc/2026/Workshop/FMSD — FMSD @ ICML 2026 SpotlightOral_

### Official Review · Reviewer_E37o · 2026-05-18
**the proposed method is simple and effective, and is well-justified by the experiments in low-data regime**

**Rating:** 9
**Confidence:** 4

**Review:**

The paper proposes a simple and effective detection method for AI-generated images under low-data constraints, which reduce image features into low dimensions (e.g., via PCA) and apply TabPFN to do classification. It's an interesting method and the experiments show good improvements over the baselines and justified the paper's motivation, especially when the available training samples are scarce (e.g., k=25), and it would be also good to see how different dimension reduction methods affect the detection results, as already mentioned by the authors in the future work section.

---

### Official Review · Reviewer_DMmC · 2026-05-20

**Rating:** 6
**Confidence:** 4

**Review:**

- Summary: This paper proposed a straightforward idea for low-data AI -generated image detection. Each image is encoded by a frozen DINOv3 into a 768 dimensional CLS feature, reduced to 500 dimensional by PCA, and classified by Tab PFN using in-context tabular inference rathan than training a task specific classifier. The method is evaluated on GenImage across pooled, per generator, single generator transfer, and pairwise generator transfer protocols.

- Strengths: The method replaces data specific training with in-context adaptation using a structured data foundation model. In practice, the image encoder is fixed, and the adaptation only required changing the labeled TabPFN. The paper is also honest about the trade-off with LATTE. It positions TabPFN as a complementary low-data adaptation mechanism.

- Weaknesses and Questions: The method (forzen visual features+PCA+TabPFN) is essential but the novelty is mostly in the application and evaluation framing. Also ablation analysis are needed to show that the gain is specifically due to TabPFN's structured in-context prior rather than simply a strong classical classifier on good DINO features. Also why is LATTE only evaluated at k=150,300,625 while TabPFN uses k=25?

---

### Official Review · Reviewer_AfrX · 2026-05-22
**Elegant idea, methodology section should be extended**

**Rating:** 7
**Confidence:** 3

**Review:**

### Summary
The paper proposes a simple in-context learning algorithm based on the foundation model TabPFN for detecting synthetic images. Features are extracted from images using DINOv3, projected via PCA, and finally predicted in a binary setting with TabPFN. The approach works especially well in limited data settings.

### Strengths
- The method is elegant, simple, and easily applicable to new datasets. As it uses TabPFN and does not require its own training pipeline, it is a great fit for this workshop.
- The empirical evaluation provides interesting insights. The model is evaluated on different generators using different metrics. Furthermore, features are ablated, and a simple MLP baseline is included. The results demonstrate the effectiveness of the proposed approach.

### Weaknesses
- The methodology section is too concise. As the approach consists of only a few steps, the authors should provide more detail. The lines L125-L135 (left) remain mostly unclear, even though they form the majority of the approach.
- The plots are hard to read. I recommend increasing the font sizes and fixing the spacing of the x-ticks-
- While the different components are well ablated, the authors should extend the comparison to LATTE and other SOTA baselines.